# Fabrication of Electrospun Cellulose Acetate/Nanoclay Composites for Pollutant Removal

**DOI:** 10.3390/polym14235070

**Published:** 2022-11-22

**Authors:** Petya Tsekova, Olya Stoilova

**Affiliations:** Laboratory of Bioactive Polymers, Institute of Polymers, Bulgarian Academy of Sciences, Akad. G. Bonchev St, bl. 103A, 1113 Sofia, Bulgaria

**Keywords:** electrospinning, cellulose acetate, nanoclays, Cr(VI) adsorption, methylene blue removal

## Abstract

The creation of eco-friendly clay-based composites for pollutant removal by adsorption still remains a challenge. This problem might be successfully solved by the development of electrospun polymer–clay composites. For the first time in this study, a one-step fabrication of cellulose acetate (CA) fibers filled with commercially available nanoclays (NCs) was described. The optimal ratio at which CA/NCs dispersions remained stable was accomplished by varying the nanoclay concentration with respect to CA. Furthermore, the selected solvent system and the electrospinning conditions allowed for the successful fabrication of electrospun CA/NC composites. It was found that the composites’ surface morphology was not affected by the incorporated nanoclays and was the same as that of the electrospun CA fibers. The performed analyses clearly showed that CA and nanoclays did not react during the electrospinning process. It was found that the distribution of nanoclay layers probably was a mixture of intercalated and exfoliated structures. Notably, the type of the nanoclay strongly influenced the adsorption ability of CA/NC composites toward Cr(VI) ions and MB dye. These results suggested that the fabricated CA/NC composites are suitable for pollutant removal due to their specific structure.

## 1. Introduction

Water pollution by heavy metal ions and dyes has become a major global problem, threatening the environment and living organisms [1,2]. Various adsorbents such as activated carbon, zeolites, and clay minerals have been successfully applied in water treatment [3,4,5,6,7,8,9,10]. Among them, clay minerals have shown promising results in water remediation because of their specific layered structure, in addition to their abundance in nature and low cost [11,12,13,14]. Moreover, a net negative charge on the structure of these layered silicate minerals is neutralized by the adsorption of positively charged heavy metal ions or dyes, thus determining their high adsorption capabilities [15,16]. Despite all of the aforementioned beneficial characteristics of clays, their separation after the adsorption process continues to be a major drawback. It was shown that this problem can be solved by the development of polymer–clay composites. As a result of the extensive research in this field, a wide range of clay-based composites using polymers have been developed [17,18,19,20,21,22,23,24,25,26,27]. Thus, by combining the properties of the fillers and polymers, materials with improved properties have been created. However, the creation of eco-friendly composites in which the clays are homogeneously dispersed and with no phase separation still remains a challenge.

Electrospinning has drawn a lot of attention as the only technique for the simple and effective fabrication of polymer fibers [28]. High surface area, high porosity, and light weight are some of the most significant advantages of the electrospun polymer materials in water treatment [29,30,31,32,33]. Therefore, the incorporation of clays into electrospun polymer fibers is a suitable strategy to modulate adsorption properties in composites [34,35]. Cellulose acetate (CA) is a biocompatible and biodegradable ester of the most abundant polysaccharide in nature–cellulose. In contrast to cellulose, CA has good solubility in organic solvents, consequently making it an excellent polymer for electrospinning [36,37,38,39,40]. Currently, CA is one of the most commonly used polymers in the preparation of electrospun membranes with a good prospect of application in filtration, separation, and water treatment [41,42,43,44]. However, the studies on obtaining cellulose acetate/clay-based composites by electrospinning for pollutant removal are still scarce [45,46]. In this regard, in the present study a one-step preparation of CA fibers filled with nanoclays was described. For the first time, commercially available nanoclay–Nanomer^®^ PGV, Nanomer^®^ 1.28E, and Nanomer^®^ 1.31PS were successfully dispersed into CA solution and then electrospun. The selected solvent system (acetone/water = 80/20 *v*/*v*) and CA/nanoclays ratio allowed the preparation of stable CA/nanoclay mixtures and their electrospinning. The surface morphology of the electrospun composites was observed by scanning electron microscopy (SEM). Further, electrospun composites were characterized in detail by ATR-FTIR spectroscopy, X-ray diffraction (XRD), and thermal gravimetric analyses (TGA). Considering the potential application for pollutant removal, the adsorption ability of electrospun CA/nanoclay composites in water was evaluated toward Cr(VI) ions and methylene blue (MB) as a model pollutant dye, compared with this of electrospun CA.

## 2. Materials and Methods

### 2.1. Materials

Cellulose acetate (CA, 30,000 g/mol and acetyl content 39.8%) and three different nanoclays (NCs)—hydrophilic bentonite (NC1, Nanomer^®^ PGV), montmorillonite clay modified with 25–30 wt% trimethyl stearyl ammonium (NC2, Nanomer^®^ 1.28E), and montmorillonite clay surface modified with 15–35 wt% octadecylamine and 0.5–5 wt% aminopropyltriethoxysilane (NC3, Nanomer^®^ 1.31PS)—were purchased from Sigma-Aldrich (St. Louis, MO, USA) and used as received. Acetone (≥99.5%), methylene blue B (MB), and potassium dichromate (K_2_Cr_2_O_7_) were supplied by Merck (Darmstadt, Germany). All chemicals were of analytical grade and used without further purification.

### 2.2. Electrospun Composites Fabrication

Fibrous CA/NC composites were fabricated by electrospinning. First, CA was dissolved in a mixed-acetone/H_2_O solvent = 80/20 *v*/*v* at a concentration of 10 wt%. Then, three different composites were prepared by adding the respective nanoclays (10 wt% with respect to CA) to the CA spinning solution. The obtained CA/NC1, CA/NC2, and CA/NC3 mixtures were homogenized by vigorous stirring and sonication (Bandelin Sonorex, 160/640 W, 35 kHz, Berlin, Germany) for 1 h. Then, mixtures were loaded into a 10 mL Luer-Lock tip plastic syringes capped with a 20-gauge needle, placed horizontally in syringe pumps (NE-300, New Era Pump Systems, Inc., Farmingdale, NY, USA), and delivered at a constant feed rate of 3 mL/h. The electrospinning of the mixtures was conducted at an applied voltage of 25 kV, a tip-to-collector distance of 15 cm, and a collector rotation speed of 1000 rpm. Electrospun CA/NC composites were dried under vacuum at 30 °C in order to remove any solvent residues.

### 2.3. Composites Characterization

The surface morphology of different CA/NC composites was observed by scanning electron microscopy (SEM, Philips 515, Tokyo, Japan). Specimens were placed on the sample holders and vacuum-coated with carbon. Attenuated total reflection–Flourier-transformed infrared spectroscopic (ATR–FTIR) analysis was performed using an IRAffinity−1 spectrophotometer (Shimadzu Co., Kyoto, Japan) equipped with a MIRacle^®^ ATR accessory (PIKE Technologies, Fitchburg, WI, USA). All spectra were corrected for H_2_O and CO_2_ using IRsolution internal software.

X-ray diffraction (XRD) analyses were performed using a D8 Bruker Advance (Billerica, MA, USA) diffractometer with filtered Cu Kα radiation and a LynxEye detector at room temperature. XRD spectra were recorded in the 2θ range from 5° to 70° with a step size of 0.02° and counting time of 1 s/step. Phase identification was performed using the Diffracplus EVA using ICDD-PDF2 Database. Thermal gravimetric analyses (TGA) were carried out on Perkin Elmer TGA 4000 (Waltham, MA, USA) at a heating rate of 10 °C/min and an argon flow of 60 mL/min. Instrument control, data acquisition, and data processing were performed by Pyris v.11.0.0.0449 software.

### 2.4. Adsorption Studies

Cr(VI) adsorption from aqueous solutions by the electrospun CA/NC composites was conducted as follows. A stock solution of K_2_Cr_2_O_7_ (3.4 × 10^−4^ mol/L) in distilled water was prepared. Then, 9 mg of CA/NC1, CA/NC2, and CA/NC3 composites was cut, weighed, and put into 10 mL Cr(VI)-containing solution in three different reagent bottles. The experiments were carried out in a thermostatic shaker bath at 25 °C and for 24 h, in order to ensure that the adsorption had reached equilibrium. Finally, the initial and final concentrations of chromium were determined using flame atomic absorption spectrometry (Thermo SOLAAR M5, Thermo Fisher Scientific, Waltham, MA, USA) at a wavelength λ = 357.9 nm and an air/acetylene gas mixture. The adsorption capacity (Q, mg/g) and the removal efficiency (RE, %) of the CA/NC composites were calculated from the following equations:Q = (C_0_ − C_e_) × V/m,(1)
RE = (C_0_ − C_e_)/C_0_ × 100,(2)
where C_0_ and C_e_ are the initial and the equilibrium Cr(VI) concentrations, V is the volume of the Cr(VI) solution, and the m is the dry mass of the CA/NC composites.

In addition, the adsorption of methylene blue B (MB) by the electrospun CA/NC composites was also evaluated. For that purpose, 9 mg of CA/NC1, CA/NC2, and CA/NC3 composites was cut, weighed, and immersed into 10 mL of MB aqueous solution (2.0 × 10^−5^ mol/L) and kept for 1 h in the dark to attain equilibrium. Then, samples were shaken (80 rpm) in a thermostatic shaker bath at 25 °C for 48 h. MB aliquots were withdrawn at different adsorption times, and their absorbance was recorded at a wavelength of 660 nm using UV–vis spectrophotometer (DU800, Beckman Coulter Inc., Brea, CA, USA). The RE (%) of the CA/NC composites was calculated by Equation (2). In both adsorption experiments, the specific surface area of the composite samples was 2.5 m^2^/g.

## 3. Results and Discussion

In our previous study, a facile and effective approach was developed for the fabrication of uniform and defect-free CA fibers in a mixed-acetone/water-solvent system [47]. In this study, the focus was on the fabrication of CA fibers filled with nanoclays and the study of their effect on the adsorption properties of the obtained CA/NC composites with respect to their targeted application. For that reason, three different nanoclays were used: hydrophilic bentonite (NC1) and organo-modified montmorillonite clays, i.e., modified with trimethyl stearyl ammonium (NC2), and octadecylamine and aminopropyltriethoxysilane (NC3), respectively. Preliminary experiments showed that all nanoclays dispersed well in the CA solution. In addition, by varying the nanoclay concentrations from 5 to 15 wt% with respect to CA, the optimum at which the CA/NC dispersions remained stable was found. The selected solvent system (acetone/water = 80/20 *v*/*v*) and CA-to-nanoclay ratio allowed for the preparation of stable CA/NC mixtures and their electrospinning. Thereby, by varying the electrospinning conditions (applied voltage from 20 to 30 kV, needle tip-to-collector distance from 10 to 20 cm, and flow rates from 1.5 to 3 mL/h), fibrous CA composites containing NC1, NC2, and NC3 were successfully fabricated.

### 3.1. Morphological and Structural Characterisation of the Fabricated CA/NC Composites

It was found that the selected electrospinning conditions (25 kV applied voltage, 15 cm needle tip-to-collector distance, and 3 mL/h flow rate) and concentrations were suitable for the fabrication of uniform and defect-free composite CA fibers containing 10% nanoclays. The surface morphology of the electrospun CA (Figure 1a) and CA/NC composites (Figure 1b–d) was observed by SEM. SEM images indicated that characteristic flat-ribbon-shaped fibers were obtained by the electrospinning of CA in a mixed-acetone/water-solvent system [47,48]. As seen, the fibrous CA/NC composites contain fibers with a smooth surface on which no nanoclays are observed (Figure 1b–d). In addition, the composite fibers had the same surface morphology as neat electrospun CA fibers. Therefore, three different nanoclays were successfully incorporated into CA fibers in one step by electrospinning, and their addition had no significant effect on the fiber’s surface morphology.

The FT−IR spectra of the CA/NC composites were compared with the spectra of neat electrospun CA (Figure 2) and pristine nanoclays (Appendix A). The CA spectrum showed the characteristic absorption signals, with the carbonyl stretching band at 1738 cm^−1^, methyl bending at 1368 cm^−1^, CH_3_–C = O at 1225 cm^−1,^ and the stretching C–O band at 1036 cm^−1^. The FT–IR spectra of pristine nanoclays (Appendix A) revealed the vibrations of Si–O stretching at 980 cm^−1^ (NC1), 995 cm^−1^ (NC2), and 997 cm^−1^ (NC3); –OH bending at 1636 cm^−1^ (NC1), 1643 cm^−1^ (NC2), and 1614 cm^−1^ (NC3); and –OH and Al–Al–OH stretching at 3613 cm^−1^ (NC1), 3618 cm^−1^ (NC2), and 3624 cm^−1^ (NC3), which is in good agreement with the literature [49,50]. Apparently, the FT-IR spectra of electrospun CA/NC composites (Figure 2) include the major characteristic peaks of CA. It should be noted that there is no sufficient indication of the presence of nanoclays in the CA fibers because of the overlapping of the characteristic peaks. However, the absence of shifts of the bands again proves that CA and nanoclays did not react during the electrospinning process.

XRD analysis was performed in order to determine whether nanoclays are intercalated or exfoliated in the CA-based composites. The XRD patterns for pristine NC1 (Appendix A) displayed strong characteristic peaks at diffraction angles 6.7°, 19.8°, 28.7°, and 35°, corresponding to the (001), (110), (210), and (124) planes of the bentonite [51]. It is noteworthy that the (001) interlayer d-spacing of both organo-modified nanoclays was shifted towards lower 2θ values (Appendix A). This result demonstrated that the larger size of organic molecules probably increased the interlayer distance, exfoliated the nanoclays, and led to the formation of a disordered clay structure [52]. In addition, pristine NC2 and NC3 displayed almost the same strong characteristic peaks at diffraction angles 2θ = 19.7°, 26.6°, and 35°, corresponding to the (110), (210), and (124) planes for the montmorillonite (Appendix A). Further, XRD analysis provided information on how nanoclay lamellae were dispersed in the CA during the formation of fibrous CA/NC composites by electrospinning. The XRD patterns presented in Figure 3 show that all types of CA/NC composites displayed one broad peak, indicating the amorphous structure of CA. The disappearance of the (001) diffraction peak in the XRD patterns of CA/NC2 and CA/NC3 composites denoted that probably the CA chains diffused into the NC2 and NC3 galleries and expanded their structure. However, the CA/NC1 composite displayed a weak diffraction peak at about 2θ = 6.7°, indicating a small amount of intercalated NC1 layers. This implies that the successful incorporation of nanoclays into CA fibers decreases the intensity of their characteristic peaks in the composites. Probably in the CA/NC composites, the distribution of nanoclay layers was a mixture of intercalated and exfoliated structures [45,53].

### 3.2. Thermal Stability of the Fabricated CA/NC Composites

The presence of nanoclays in the CA fibers was further proved by the thermogravimetric analyses. The thermograms of pristine nanoclays, electrospun CA, and composites are shown in Figure 4. Apparently, unmodified NC1 bentonite nanoclay showed almost 13% weight loss in the temperature range 35–120 °C, corresponding to the loss of moisture (desorption of water). In contrast, the weight losses in the same temperature range for organo-modified NC2 and NC3 montmorillonite clays were only about 3% and 2%, respectively, which proved that NC1 is more hydrophilic than NC2 and NC3.

The second step of the thermal degradation of NC1 was around 700 °C, with a weight loss of 5% due to the dehydroxylation of the bentonite lattice [54,55]. The most distinguished difference between unmodified NC1 and both organo-modified NC2 and NC3 clays was in the temperature range 200–450 °C, where the total weight loss of about 30% for both NC2 and NC3 nanoclays occurred. This weight loss could be attributed to the degradation of respective and less thermally stable organic modifiers in NC2 and NC3 clays. The residual weights of NC1, NC2, and NC3 at 800 °C were about 81%, 64%, and 66%, respectively. Hence, after the removal of moisture and organic modifiers, nanoclays did not decompose thermally because the alumina and silicate layers are highly stable at this temperature.

Further, the thermal decomposition of electrospun CA and CA/NCs began at 300 °C and ended at 500 °C due to the decomposition of CA (Figure 4). It was found that electrospun CA and CA/NC1 composite showed only one decomposition peak at about 408 °C (TGA derivative not shown). In addition to the decomposition peak at about 415 °C, the CA/NC2 and CA/NC3 composites show a second peak between 250 and 400 °C (TGA derivative not shown), which is due to the degradation of the corresponding organic modifier. Nevertheless, it is notable that the thermal stability of the CA/NC2 and CA/NC3 composites was slightly shifted to a higher decomposition temperature in comparison to that of CA. Finally, after the thermal decomposition of the CA/NC1, CA/NC2, and CA/NC3 composites at 800 °C, the residues of ca. 17%, 12%, and 14% appeared. These residues are caused by the inorganic components in nanoclays, which are thermally stable at this temperature. Moreover, these values were very close to the initial NC1, NC2, and NC3 amounts added to the CA solutions. Thus, in confirmation of the FT–IR results, TG analyses clearly showed that during the electrospinning CA and nanoclays did not react.

Hence, the optimal conditions for the successful incorporation of 10% of different nanoclays into CA fibers by one-step electrospinning were found. The performed analyses clearly showed that the nanoclays were successfully loaded into the CA fibers and that during the electrospinning the CA and nanoclays did not react as well.

### 3.3. Evaluation of Metal Ions and Dyes Removal

It is well known that clays are utilized as adsorbents for the removal of heavy metals and dyes from the wastewater [11,12,13,14,15,16]. Notably, clays exhibited complex adsorption mechanisms—physical by electrostatic interaction and chemical by coordination chelation and/or ion exchange [32]. According to this, and in view of the potential application of pollutant removal, the adsorption ability of electrospun CA/NC composites in water was evaluated toward Cr(VI) ions and methylene blue (MB) as a model pollutant dye and compared with this of electrospun CA.

In order to study the effect of different types of nanoclays on the adsorption properties of electrospun composites, the adsorption capacity (Q) and removal efficiency (RE) of Cr(VI) were calculated and compared (Figure 5). As expected, the reduction of the Cr(VI) ions in water occurred in the presence of composites. From the presented results (Figure 5), it is evident that CA/NC3 composite exhibited the highest Cr(VI) uptake. Obviously, the type of the embedded into CA fibers nanoclay strongly affected the adsorption ability of the electrospun composites. As seen, the amounts of Cr(VI) adsorbed onto the CA/NC1 and CA/NC2 composites were 1.4 and 3.7 times more than onto the electrospun CA. Bearing in mind that the Cr(VI) adsorption is mainly achieved by the electrostatic/ionic interactions and oxygen-containing functional groups on the composites surface, it might be assumed that the enhanced adsorption was due to the combined effect of the electrospun specific surface area of the composites and type of the loaded nanoclay. Moreover, nanoclays possess a certain cation exchange capacity, which further improves the removal efficiency of Cr (VI) ions in water from the composites.

Furthermore, the removal efficiency of the composites toward MB was also studied and monitored spectrophotometrically at 660 nm. It was distinctly seen that even after 1 h in the presence of electrospun CA and composites, a decrease in MB absorption and discoloration occurred (Figure 6a). Notably, the MB uptake was significantly increased in the presence of CA/NC1 and CA/NC3 composites (Figure 6a). In addition, the removal efficiency of MB in the presence of electrospun CA and composites was calculated. After 6 h in the presence of electrospun CA, the MB absorption reached saturation. The calculated RE for electrospun CA after 48 h was only 34% (Figure 6b). In contrast, electrospun CA/NC composites exhibited higher RE and after 48 h were 60% for CA/NC2 and almost 100% for CA/NC1 and CA/NC3 composites.

In particular, oxygen-containing functional groups in CA and the specific structure of nanoclays enable the formation of covalent bonding, electrostatic attraction, and cation exchange with MB dye. Hence, the MB sorption driving force was the electrostatic/ionic interactions between positively charged MB with negatively charged hydroxyl groups, as well electrostatic interaction and cation exchange with nanoclays. Therefore, the type of the nanoclay strongly affected the adsorption ability of CA/NC composites toward MB dye. The obtained results prove that the incorporation of nanoclays into CA fibers by electrospinning is a promising strategy that enables the fabrication of effective composites with potential for pollutant removal.

## 4. Conclusions

Novel CA/nanoclay composites were fabricated by one-pot electrospinning. The composites’ surface morphology was not affected even after successful incorporation of 10% of nanoclay into the CA fibers. Moreover, FT-IR and TG analyses proved that during the electrospinning process, CA and nanoclays did not react. Based on XRD analysis, it was concluded that the distribution of nanoclay layers in the CA/NC composites was a mixture of intercalated and exfoliated structures. The adsorption studies clearly showed that the nanoclay type strongly influenced the adsorption capability of CA/nanoclay composites toward Cr(VI) ions and MB dye. Furthermore, it was found that the CA/NC3 composite exhibited the highest Cr(VI) uptake, while the amounts of Cr(VI) adsorbed onto the CA/NC1 and CA/NC2 composites were 1.4 and 3.7 times more than onto the electrospun CA. In addition, electrospun CA/NC composites exhibited higher removal efficiency and after 48 h were 60% for CA/NC2 and almost 100% for CA/NC1 and CA/NC3 composites, respectively. These results suggest that the obtained novel electrospun composites are potential candidates for pollutant removal from water.

## Figures and Tables

**Figure 1 polymers-14-05070-f001:**
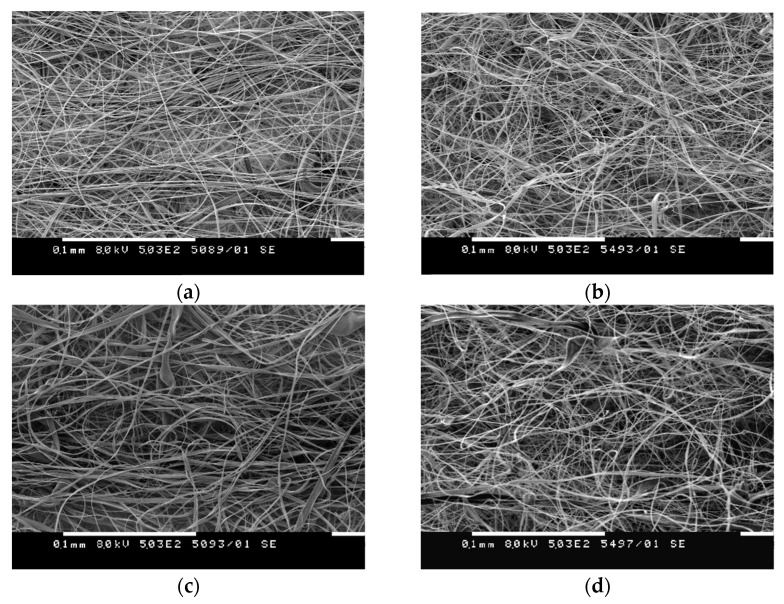
SEM micrographs of electrospun CA (**a**) and of CA/NC1 (**b**), CA/NC2 (**c**), and CA/NC3 (**d**) composites.

**Figure 2 polymers-14-05070-f002:**
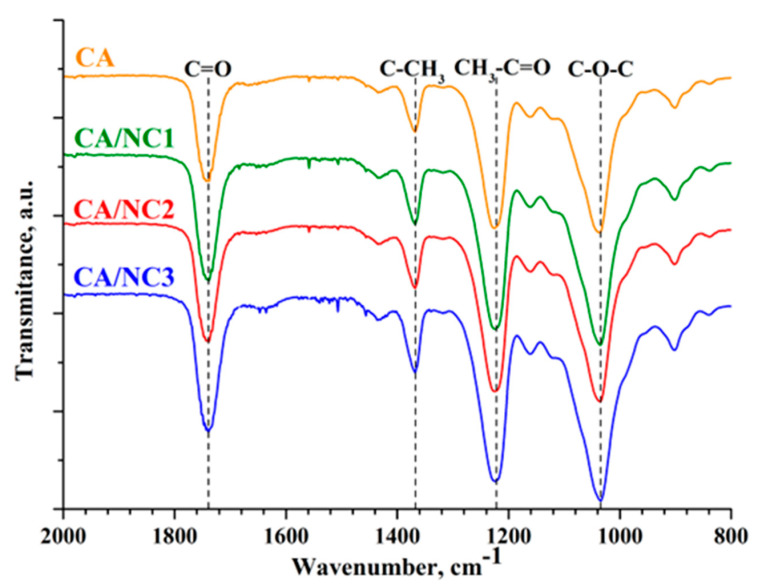
ATR–FTIR spectra of electrospun CA and of CA/NC1, CA/NC2, and CA/NC3 composites.

**Figure 3 polymers-14-05070-f003:**
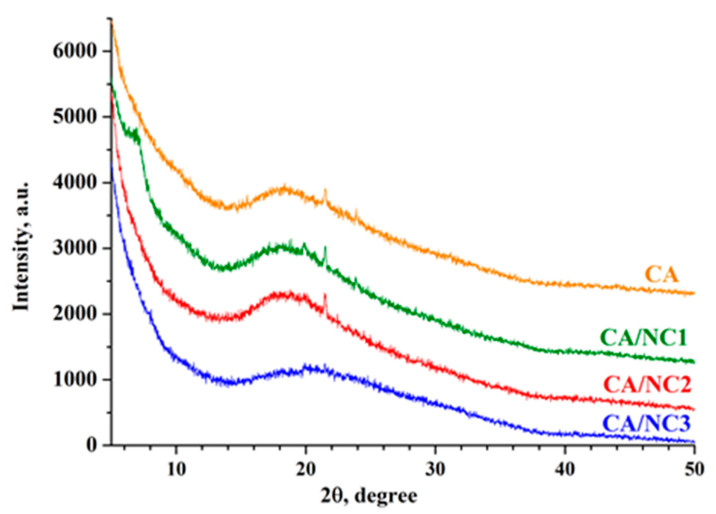
XRD patterns of electrospun CA and of CA/NC1, CA/NC2, and CA/NC3 composites.

**Figure 4 polymers-14-05070-f004:**
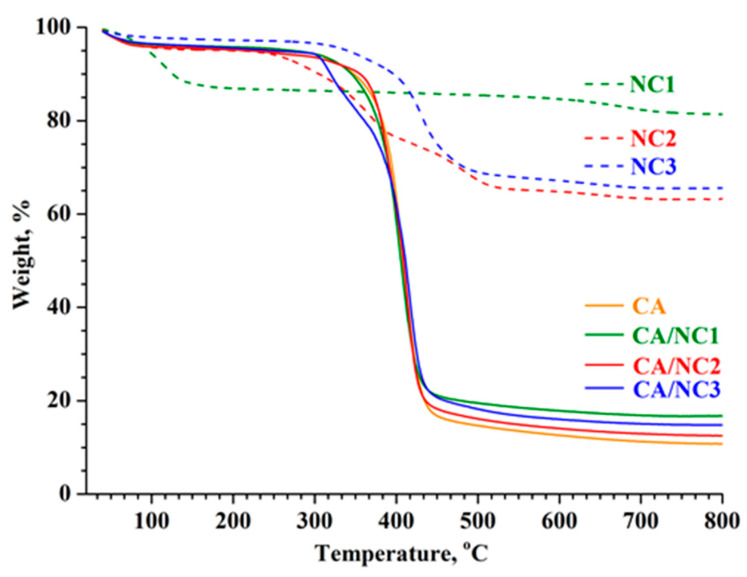
TG thermograms of pristine nanoclays and electrospun CA and of CA/NC1, CA/NC2, and CA/NC3 composites.

**Figure 5 polymers-14-05070-f005:**
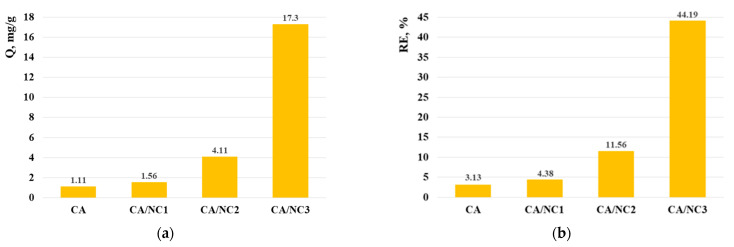
Adsorption capacity (**a**) and removal efficiency (**b**) of Cr(VI) from the electrospun CA and of CA/NC1, CA/NC2, and CA/NC3 composites (pH 5, 25 °C).

**Figure 6 polymers-14-05070-f006:**
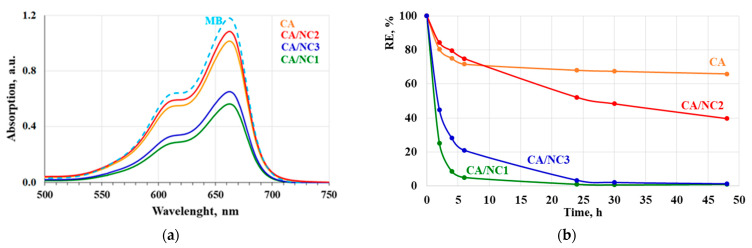
Absorption spectra of MB after 1 h (**a**) and removal efficiency of MB for 48 h (**b**) in the presence of electrospun CA and of CA/NC1, CA/NC2, and CA/NC3 composites.

## Data Availability

The data presented in this study are available on request from the corresponding author.

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
