# Peer review of "Fabrication of Electrospun Cellulose Acetate/Nanoclay Composites for Pollutant Removal"

_polymers, 2022, doi:10.3390/polym14235070_

Round 1

Reviewer 1 Report

In this manuscript, P. Tsekova et al. described the “Fabrication of electrospun cellulose acetate/nanoclays composites for pollutants removal” in detail. The samples were characterized for the removal of Cr and MB. Although, all the samples showed excellent activities, yet there are still some problems which need to be addressed. I therefore, suggest minor review at this stage based on the following questions.

1) What is the admissible concentration of different heavy metals.

2) Although, English language is good, yet there are still many mistakes which need to be addressed in the revised manuscript.

3) What is the specific surface area of the composite samples.

4) What is the effect of different ions in water on the adsorption of Cr(VI) ions and methylene blue (MB).

5) What the effect of pH on the adsorption of Cr(VI) ions and methylene blue (MB).

6) What the effect of temperature on the adsorption of Cr(VI) ions and methylene blue (MB).

7) Some very important citations are missing.

i) R. Nazir, M. Khan, R. U. Rehman, S. Shujah, M. Khan, M. Ullah, A. Zada, N. Mahmood, I. Ahmad, Adsorption of selected azo dyes from an aqueous solution by activated carbon derived from Monotheca buxifolia waste seeds, Soil Water Res. 15 (2020) 166-172.

ii) M. Ullah, R. Nazir, M. Khan, W. Khan, M. Shah, S. G. Afridi, A. Zada, The effective removal of heavy metals from water by activated carbon adsorbents of Albizia lebbeck and Melia azedarach seed shells, Soil Water Res. 15 (2020) 30-37.

iii) N. Ali, A. Zada, M. Zahid, A. Ismail, M. Rafiq, A. Riaz, A. Khan, Enhanced photodegradation of methylene blue with alkaline and transition-metal ferrite nanophotocatalysts under direct sun light irradiation, J. Chin. Chem. Soc. 66 (2019) 402-408.

Author Response

POINT-TO-POINT RESPONSE TO REVIEWER 1 COMMENTS

On behalf of all the co-authors I gratefully thank the Reviewer 1 for the thorough analysis of our manuscript, as well as for the comments.

Hereafter our responses and the changes made in the manuscript according to the recommendations of the Reviewer 1 are reported. All modifications were mark with the Track Changes tool.

Response to Reviewer 1 Comments

1) What is the admissible concentration of different heavy metals.

Response 1: After Pb, Cd and Hg, chromium is the next entry in major toxic metal series. The discharge limit of chromium from industries is less than 1 mg/L. Chromium is hazardous to health when its limit in portable water exceeds 0.5 mg/L.

2) Although, English language is good, yet there are still many mistakes which need to be addressed in the revised manuscript.

Response 2: A few grammatical corrections have been done.

3) What is the specific surface area of the composite samples.

Response 3: The specific surface area of the composite samples was 2,5 m2/g.

4) What is the effect of different ions in water on the adsorption of Cr(VI) ions and methylene blue (MB). 5)What the effect of pH on the adsorption of Cr(VI) ions and methylene blue (MB). 6) What the effect of temperature on the adsorption of Cr(VI) ions and methylene blue (MB).

Responses 4, 5, 6: The focus of the manuscript was on the fabrication of composite fibers. For that reason, we did not study in details the effect of factors such as the presence of other ions in water, pH and temperature on the adsorption of Cr(VI) ions and MB (cationic dye). In general, the effect of pH depends on the ions present in the reaction mixture as well as on electrostatic interactions with the adsorption surface. It is well-known that at low pH Cr(VI) adsorption increases, while at high pH Cr(VI) adsorption decreases. In contrast, the uptake of the MB increased with increasing pH. In addition, the sorption capacity of Cr(VI) and MB in water increased on increasing the temperature. Nevertheless, a more detailed effect of these important factors on the adsorption of Cr(VI) ions and MB will be the topic of a forthcoming study.

7) Some very important citations are missing.

  1. i) R. Nazir, M. Khan, R. U. Rehman, S. Shujah, M. Khan, M. Ullah, A. Zada, N. Mahmood, I. Ahmad, Adsorption of selected azo dyes from an aqueous solution by activated carbon derived from Monotheca buxifolia waste seeds, Soil Water Res. 15 (2020) 166-172.
  2. ii) M. Ullah, R. Nazir, M. Khan, W. Khan, M. Shah, S. G. Afridi, A. Zada, The effective removal of heavy metals from water by activated carbon adsorbents of Albizia lebbeck and Melia azedarach seed shells, Soil Water Res. 15 (2020) 30-37.

iii) N. Ali, A. Zada, M. Zahid, A. Ismail, M. Rafiq, A. Riaz, A. Khan, Enhanced photodegradation of methylene blue with alkaline and transition-metal ferrite nanophotocatalysts under direct sun light irradiation, J. Chin. Chem. Soc. 66 (2019) 402-408.

Response 7: Numerous works of true merit have appeared in the field of pollutants removal by adsorption. Among them, the above-mentioned references are valuable, but is not related to the scope of the present paper.

Reviewer 2 Report

The manuscript (Fabrication of electrospun cellulose acetate/nanoclays composites for pollutants removal ) is interesting.

However, there are some remarks have to be considered before recommending its acceptance for publication.

1- The derivative dTG has to be mentioned in this manuscript to enrich this work by monitoring more information about the thermal decomposition of the investigated constituents.

2- The Conclusion section has to be rewritten with more details.

Author Response

POINT-TO-POINT RESPONSE TO REVIEWER 2 COMMENTS

On behalf of all the co-authors I gratefully thank the Reviewer 2 for the thorough analysis of our manuscript, as well as for the comments.

Hereafter our responses and the changes made in the manuscript according to the recommendations of the Reviewer 2 are reported. All modifications were mark with the Track Changes tool.

Response to Reviewer 2 Comments

On behalf of all the co-authors I gratefully thank the Reviewer 2 for the thorough analysis of our manuscript, as well as for the comments.

Hereafter our responses and the changes made in the manuscript according to the recommendations of the Reviewer 2 are reported. All modifications were mark with the Track Changes tool.

1- The derivative dTG has to be mentioned in this manuscript to enrich this work by monitoring more information about the thermal decomposition of the investigated constituents.

Response 1: Indeed, derivative thermogravimetric (DTG) curves will give a plot of the rate of change of mass with respect to temperature against temperature. Nevertheless, these changes are explained in details in Section 3.2. and thus, including additional DTG curves will useless. However, according to the referee comment, we could support these data in Section Data Availability Statement.

2- The Conclusion section has to be rewritten with more details.

Response 2: As suggested by Referee 2 Conclusions has been rewritten with more details.
